# Mechanism of Action and Interaction of Garlic Extract and Established Therapeutics in Prostate Cancer

**DOI:** 10.3390/ijms26041777

**Published:** 2025-02-19

**Authors:** Marco Hoffmann, Jana Sauer, Marie Book, Thomas Frank Ermler, Petra Fischer, Sven Gerlach, Kareem Beltagi, Agnieszka Morgenroth, Radu Alexa, Jennifer Kranz, Matthias Saar

**Affiliations:** 1Department of Urology and Pediatric Urology, University Hospital RWTH Aachen, 52074 Aachen, Germanytermler@ukaachen.de (T.F.E.); kbeltagi@ukaachen.de (K.B.); ralexa@ukaachen.de (R.A.);; 2Center for Integrated Oncology (CIO), University Hospital RWTH Aachen, 52074 Aachen, Germany; pfischer@ukaachen.de; 3Department of Otorhinolaryngology-Head and Neck Surgery, University Hospital RWTH Aachen, 52074 Aachen, Germany; 4Department of Nuclear Medicine, University Hospital RWTH Aachen, 52074 Aachen, Germany; amorgenroth@ukaachen.de

**Keywords:** garlic, prostate cancer, natural products, drug-interaction, androgen receptor, CYP450

## Abstract

A detailed characterization of the mechanism of action of garlic extract (GE) on prostate cancer (PCa) cells is essential to ensure its safe use as a complementary therapy, particularly when combined with established treatments. A case report highlighted the potential benefits of GE in PCa management. A patient diagnosed with PCa, presenting an initial prostate-specific antigen (PSA) of 11.8 ng/mL, maintained PSA levels between 3.5 and 6 ng/mL for over 14 years with daily GE intake. To study GE’s anti-proliferative effects and interactions with established therapeutics, healthy prostate epithelial cells (PNT2) and PCa cells (LNCaP, PC3, VCaP) were treated with GE. Proliferation, Integrin β1 pattern, DNA-damage, as well as androgen receptor (AR) and Cytochrome P450 (CYP450) expression were investigated. GE reduced the proliferation of LNCaP and PC3 cells compared to healthy PNT2 cells but had contrary effects on VCaP cells. The combination of GE with standard therapies, including chemotherapy, androgen deprivation therapy (ADT), and Poly-(ADP-ribose)-Polymerase inhibitors (PARPi), reduced the efficacy of these treatments in tumor cells, potentially due to the GE-induced upregulation of the metabolic enzyme CYP2C9 in PCa cell lines. These findings indicate that while GE has anti-proliferative effects, the use of highly concentrated natural extracts must be carefully assessed by expert physicians on a case-by-case basis, especially when combined with established therapies.

## 1. Introduction

PCa is the fourth most common cancer worldwide and, with nearly 400,000 annual deaths, ranks among the ten deadliest cancers. Nearly one in three men worldwide will develop PCa, making it, alongside breast and lung cancer, one of the most burdensome cancers for healthcare systems [1]. In early PCa stages, active surveillance may suffice [2], while curative therapy is initiated upon disease progression. Treatment strategies are then tailored to the patient’s tumor characteristics and may include surgery, chemotherapy, radiotherapy, ADT, molecularly targeted therapies, such as PARPi, or combinations thereof [3,4,5,6,7]. Particularly, in systemic therapies for advanced PCa, side effects, such as nausea, anemia, and fatigue are common, while complete remission remains rare [8,9,10,11]. For this reason, the combination of established therapies with natural substances has been explored to identify new therapeutic options [12]. In this context, one focus is on how garlic interacts with established PCa therapies. Garlic has attracted scientific interest due to its broad medical properties, including anti-microbial, anti-viral, and anti-cancer effects [13,14,15,16,17,18]. Garlic’s anti-cancer effects are primarily attributed to organosulfur compounds (OSCs) derived from allicin. In general, allicin is catalyzed from alliin through the enzyme alliinase which is activated as part of the plant’s defense response after injury. By the further degradation of allicin, its associated OSCs are formed. These allicin-derived OSCs within garlic have been shown to inhibit the activation and expression of proteins that stimulate cell growth, thus targeting several cancer hallmarks as described by Hanahan and Weinberg et al. and others [19,20,21,22]. Additionally, OSCs are believed to affect the cellular redox balance. For example, when cysteinyl S-conjugates in OSCs are activated via β-lyase reactions, reactive persulfide or sulfane sulfur species are generated. These species may then interact with cysteine residues on redox-sensitive proteins, including those involved in cellular signaling [23]. Evidence suggests that S-allylcysteine, a compound found in garlic, can induce apoptosis similarly to allicin, through both caspase-dependent [24] and independent [25] mechanisms in various cancer cell types, while also reducing the toxicity of chemotherapy agents [26]. Degradation products of allicin, such as diallyl disulfide (DADS) and diallyl trisulfide (DATS), are also known to possess anti-cancer activity via apoptosis and cell cycle arrest [27,28].

In addition to that, the potential efficacy of GE was demonstrated in the case of a PCa patient who opted for daily GE consumption instead of conventional treatment, resulting in a significant reduction in PSA levels (see Appendix A). This case makes GE a potential candidate for a complementary PCa therapy and has spurred further investigation into the effects on PCa cell lines to quantify its impact and underlying mechanisms. Despite the promising prospects of such cases, a comprehensive assessment of natural substances as complementary therapies in combination with established treatments remains essential [29]. Drug metabolism plays a crucial role in therapeutic efficacy and is largely governed by cytochrome P450 (CYP) enzymes [30]. CYP2C9, for instance, is responsible for metabolizing 15 to 25% of all drugs [31]. Genetic polymorphisms and interactions with various substances can affect CYP-activity, resulting in variations in drug metabolism [32]. Herbal supplements, such as GE, can alter drug metabolism through herb–drug interactions (HDIs) [33]. Therefore, integrating complementary therapies like GE with conventional treatments requires a detailed understanding of their pharmacological interactions to ensure both safety and efficacy. Given the widespread use of herbal supplements among cancer patients [34,35], detailed individual characterizations of possible HDIs are crucial to optimize existing therapeutic strategies and minimize adverse effects [29,36,37]. In conclusion, while GE shows promise as a complementary therapy for PCa, its interactions with established therapeutic agents necessitate careful consideration for each patient. This article aims to elucidate the mechanism of action of GE and its interactions with established PCa therapeutics to facilitate the safe integration of GE into PCa treatment regimens and raise awareness of this critical issue among healthcare professionals and researchers.

## 2. Results

### 2.1. Effect of GE on the Proliferation of Healthy Prostate Epithelial and PCa Cell Lines

To identify the most effective dose of GE for treating PCa cells, a concentration series (GE1, GE2, GE3) was incubated on the PCa cell line LNCaP. To validate potential selective effects on PCa cells, healthy prostate epithelial cells PNT2 were also treated to quantify the effect on proliferation by MTS (Figure 1). PNT2 cells showed significant reductions in proliferation for all three concentrations (GE1-GE3) compared to controls after 6 h (h) treatment (Figure 1A). In LNCaP cells, a dose-dependent reduction in proliferation was demonstrated with significant reductions for GE2 at all time points (Figure 1B). To evaluate the effect of this specific GE concentration (GE2) on additional PCa cell lines, PC3 and VCaP cells were treated with GE and compared to controls. In PC3 cells, a significant reduction in proliferation was detected after 2 h of treatment (Figure 1C). In contrast, VCaP cells exhibited an opposite effect showing enhanced proliferation with GE2 treatment (Figure 1D). Moreover, live/dead-staining showed no significant reduction in cell-viability for VCaP cells following GE2 treatment (Figure 1E). When comparing the response of PCa cells to healthy PNT2 cells, a significant reduction in proliferation was only observed in LNCaP cells with GE2, while PC3 cells showed a non-significant reduction, and VCaP cells exhibited a significant increase in proliferation (Figure 1F).

### 2.2. Mechanisms of Action of GE on PCa Cells

The potential mechanisms of action of GE on PCa cells were characterized subsequently. DNA-damage induced by GE was quantified using a Comet assay. After 2 h of treatment with GE2, a significant increase in DNA-damaged cells was observed in all three PCa cell lines, as demonstrated by microscopy images of Comet assays following DNA staining (Figure 2A–C). Quantification of DNA-damaged cells showed a significant increase in all three PCa cells lines compared to control samples (Figure 2D). Cell cycle analysis revealed that, despite DNA damage, VCaP cells remain in the G1 phase after GE-treatment at higher levels compared to all other cell types tested in this study (Appendix A).

A characterization of Integrin β1 was performed using LNCaP cells with significant GE inhibition compared to control samples by immunefluorescence (Figure 3A,B). A localization study between control cells (Figure 3C) and GE-treated cells (Figure 3D) quantified this signal shift and revealed a significant reduction in membrane signals by GE treatment in LNCaP cells (Appendix A). Immunefluorescence studies between untreated tumor cells (Figure 3E) and GE-treated cells (Figure 3F) revealed a twofold and significantly increased Caspase-8 signal intensity (Appendix A) associated with a significant increased colocalization of Integrin β1 and Caspase-8 (white boxes in E/F and Appendix A). The anti-apoptotic protein B-cell lymphoma 2 (Bcl-2) was quantified by a Western blot after the GE treatment of LNCaP cells (Figure 3G), indicating a significant reduction by 80% compared to control samples (Figure 3H).

### 2.3. Interaction of GE with Androgen Receptors in Prostate Cancer Cells

The expression of AR after GE2 treatment was characterized by immunefluorescence (Figure 4A–C) and qRT-PCR (Figure 4E). For PCa cell lines VCaP and LNCaP, significantly reduced fluorescence intensities of AR were observed by immunefluorescence (Figure 4D). However, using qRT-PCR, no significant reduced expression levels for all cancer cells were detected (Figure 4E).

### 2.4. Combination of GE with Chemotherapy, ADT, and PARPi

A combination of GE2 and established PCa therapeutics was investigated by means of MTS with regard to possible synergistic and antagonistic effects. Since no effective reduction in proliferation could be shown in VCaP cells following GE treatment, these analyses were performed on LNCaP cells in comparison to healthy prostate epithelial cell PNT2. Increasing concentrations of Docetaxel (chemotherapy), Enzalutamide (ADT), and Olaparib (PARPi) were combined with GE2 treatment (Figure 5A,B). Docetaxel showed increased antagonistic effects with increasing concentration when combined with GE, both in healthy and PCa cells, compared to GE treatment alone (Figure 5A–C). These effects were particularly strong in LNCaP PCa cells (Figure 5C). A combination of GE and ADT demonstrated synergistic effects in healthy PNT2 cells at all concentrations (Figure 5A,C). For Olaparib, synergistic effects were observed at concentrations up to 1 µM, while additive effects were detected at 10 µM in healthy PNT2 cells (Figure 5A,C). In LNCaP PCa cells, the lowest concentrations of ADT and PARPi showed additive effects, whereas higher concentrations exhibited antagonistic effects (Figure 5D). In the case of PC3 PCa cells, combinations of GE with the most effective concentrations of Docetaxel, Enzalutamide, and Olaparib revealed synergistic effects with 10 nM Enzalutamide and GE (Figure 5D). The DNA damage induced by a combination of PARPi and GE was additionally quantified by a Comet assay. Significantly increased proportions of DNA-damaged cells were associated with reduced proliferation for LNCaP and PC3 PCa cells as a result of complementary therapy with GE compared with pure Olaparib treatment (Appendix A). The expression of the metabolic enzyme CYP2C9 was investigated to characterize the reduced efficacy of the therapeutic agents in tumor cells upon combination with GE. All tumor cells showed an altered CYP2C9 expression by GE treatment, with significantly increased expression levels for LNCaP and VCaP but non-significantly increased levels with high variations of PC3 PCa cells. A CYP2C9 activity assay for LNCaP PCa cells also showed significantly increased activities of this metabolic enzyme. In addition to these combination therapies, the uptake of 177Luthetium (^177^Lu)-PSMA-617 following GE2 treatment was characterized. Significant changes in PSMA uptake were observed in LNCaP and VCaP cells, with uptake rates of ^177^Lu-PSMA-617 reduced by up to 35% (Appendix A).

## 3. Discussion

The referenced case report shows potential effects of GE on PCa through reduced PSA levels over a period of more than 10 years. However, this individual case must be considered a preliminary observation, as the patient exhibited favorable parameters primarily due to clinical factors and overall health status. In detail, the patient’s clinical tumor parameters (Gleason 6 (3+3), 5/13 cores positive, 1CD-0-DA M 8140/3, tumor stage 2b, iPSA = 11.6), and excellent overall fitness, attributing the PSA reduction solely to GE intake should be approached with caution. Supporting imaging data, such as MRI to demonstrate tumor remission over time, are lacking and the patient’s good general health may indicate a favorable tumor prognosis independent of GE. Bolam et al. found a link between increased fitness and reduced PCa risk in nearly 57,000 Swedish men aged 30 to 50 [38]. Additional research has consistently shown a relationship between fitness level and PCa risk or progression, with a significant risk reduction correlated with high fitness levels [39,40,41].

In our study, we quantified the effects of GE using healthy prostate epithelial (PNT2) and PCa-cell lines (LNCaP, PC3, and VCaP). Pimentel et al. reported a dose-dependent reduction in the proliferation of PNT2 cells after 48 h of GE treatment [42]. The anti-proliferative effect of the GE component Diallyldisulfid (DADS) in LNCaP cells was demonstrated by Gunadharini et al. at concentrations between 25 and 100 μM [20,22]. Both allicin and DADS have shown anti-proliferative effects in the cell lines PC3 (PCa), MCF-7 (breast cancer), HT-29 (colon cancer), and Ishikawa (endometrial cancer). The effects in MCF-7 cells were associated with cell cycle arrest in the G1 and G2 phases, while in PC3 cells, they were related to the G2 phase [21]. Cells with DNA damage are typically arrested at the G2/M and G1/S phase checkpoints to prevent the further accumulation of damage [43,44].

However, VCaP cells showed increased proliferation with GE treatment. Although data on the impact of GE or its components on VCaP cells are limited, previous studies have shown tocopherols and tocotrienols, both enriched in GE, can reduce VCaP xenograft growth in mice [45]. An explanation for these opposing results might be the induction of cellular senescence (CS), which leads to persistent cell cycle arrest in the G1 phase with high metabolic activity [46]. This hypothesis was underlined by the quantification of G1 phase arrest, with the highest values for VCaP cells compared to LNCaP and PC3 induced by GE-treatment. The GE compound alliin, similar to allicin, has been shown to induce CS in MCF-7 cells after 72 h [47]. Surprisingly, the mechanisms of action of GE that align with chemotherapy, ADT, and PARPi showed reduced effectiveness in PCa cells when used as complementary therapy. This may be primarily due to a specifically altered and adapted metabolic activity in cancer cells [48], leading to increased CYP2C9 activation [49], which ultimately results in the reduced effectiveness of the therapeutics. CYP2C9 is responsible for the metabolism of approximately 15–25% of all drugs [31]. Explicit data for the CYP2C9 in PCa are not available to our knowledge. However, differential expression patterns of the CYP450 family seem to play a role in PCa [50,51]. Further analysis of additional CYP variants and their modulation by GE intake would provide valuable insights. Given the correlation between CYP2C9 regulation and herbal components observed in this and other studies [25], this interaction appears plausible, although the analysis of other CYP-variants, such as CYP3A4, CYP2D6, and CYP1A2, could provide a more detailed understanding of the relationship between GE-induced increased enzymatic activity and the metabolism of established therapeutics. Moreover, the high variability in CYP2C9 mRNA analysis for PC3 cells, which showed no significant changes in expression, could also explain the slightly improved efficacy of GE as complementary therapy for this cell type. The only additive effect observed in PC3 cells when combining GE with an androgen receptor blockade can be attributed to the ineffectiveness of enzalutamide in this AR-negative cell line, resulting in a higher relative efficacy of GE alone. PSMA-targeted therapies, such as the FDA-approved ^177^Lu-PSMA-617 radioligand therapy or innovative miRNA approaches, which use PSMA as a target receptor for miRNA delivery [52,53], may be impaired by the intake of GE, further emphasizing the importance of such HDIs.

The extent to which the effects observed in this study are reproducible and relevant for the actual in vivo function of GE remains speculative. It is important to emphasize that this study aimed to characterize the potential interactions of GE with tumor cells without intending to claim direct clinical applicability. The concentrations that demonstrated efficacy in this study are likely much higher than the bioavailable concentrations in vivo. Ideally, clinical trials would be necessary in an appropriate patient cohort to further assess these effects. Additionally, the appropriate method of administration must be considered, as some studies, for example, have used intraperitoneal (i.p.) injections in mouse models [46], whereas the supplementation of GE would more likely involve oral intake in patients.

Nevertheless, Studies indicate that 20% to 90% of cancer patients use dietary supplements or natural therapies [35]. This high prevalence, combined with patients’ reluctance to disclose such use to their physicians [54], underscores the urgent need for thorough patient histories and proactive discussions. Clinicians should consider dietary supplements as potentially active substances [46], particularly regarding their combination with established therapeutics and possible interactions that may compromise treatment outcomes.

## 4. Materials and Methods

### 4.1. Cell Culture

Three human PCa cell lines, LNCaP (Merck, Darmstadt, Germany, 89110211), PC3 (ATTC, Manassas, VA, USA, CR-1435), and VCaP (Merck, 06020201), were used. A human epithelial prostate cell line PNT2 (Merck, 95012613) was included as a healthy control. All cell lines were cultured within appropriate media (see Table 1) in a humidified atmosphere at 37 °C and 5% CO_2_. During cell growth, cultivation was performed by exchanging the medium every 2 to 3 days. Cells were subcultured at a density of 80% to 90% by incubation with 0.25% trypsin-EDTA solution (Thermo, Waltham, MA, USA, 25200056) at 37 °C for 4–10 min. For cultivation of LNCaP and VCaP, T75 flasks (VWR, Radnor, PA, USA, 734-2584) were coated with 0.01% (*v*/*v*) human fibronectin solution (FN) (Merck, FN0635) (diluted in 1× phosphate buffered saline (PBS) (Thermo, 10010023)). FN-coating was used for all cell lines during each experiment to provide the highest comparability. LNCaP (40,000 cells/cm^2^), PC3 (30,000 cells/cm^2^), VCaP (130,000 cells/cm^2^), and PNT2 (30,000 cells/cm^2^) cells were seeded on 96-well plates (VWR, 734-2327) (0.32 cm^2^) (proliferation assay MTS), on 24-well plates (VWR, 734-2325) (1.9 cm^2^) (CYP2C9 activity assay), on 12-well plates (VWR, 734-2324) (4.0 cm^2^) (Comet assay, mRNA and protein isolation), and on glass-bottom µ-dish (Ibidi GmbH, München, Germany, 81158) (3.5 cm^2^) (immunefluorescence). A hemocytometer was used for cell counting.

### 4.2. Preparation and Treatment of GE

GE isolation was performed as described previously [55]. Briefly, 350 g of garlic cloves were blended in 250 mL 40% ethanol to activate alliinase, then stored in an airtight jar at 4 °C for 5 days. The mixture was filtered, and the solution centrifuged at 4300× *g* for 10 min. The supernatant was collected and stored at −20 °C. The extract contained ~22% ethanol. Isolated GE was diluted in cell culture media to reach final concentrations of 0.1% (*v*/*v*) (GE1), 0.5% (*v*/*v*) (GE2), and 1% (*v*/*v*) (GE3). Cells were treated with GE1, GE2, and GE3 for 2 to 48 h. Ethanol controls to analyze pure GE effects were included according to ethanol concentrations in GE extract reached by the used receipt.

### 4.3. Preparation and Treatment of PCa Therapeutic Agents

Docetaxel (Selleckchem, Houston, TX, USA, S1148), Enzalutamide (Selleckchem, USD150), and Olaparib (Selleckchem, S1060) were used as established therapeutic agents to analyze combined anti-proliferative effects with GE. Agents were dissolved in 100% DMSO to yield stock solutions of 10 µM. Stock solutions were further diluted in cell culture media (*v*/*v*) to obtain final concentrations of 1 nM to 8 nM for Docetaxel, 0.01 µM to 10 µM Enzalutamide, and Olaparib. Toxic effects of DMSO could be excluded due to low concentrations (<0.01%) in cell culture media. Cells were treated with therapeutic agents for 48 h and in combination with GE, GE2 was added for the last 2 h of treatment. For radiolabeling, 332 µL of [^177^Lu]LuCl_3_ solution (100–1000 MBq) in dilute HCl (0.04 M) was added to a solution of 5.6 µg PSMA-I&T in 162 µL ammonium acetate buffer (pH 4.4) within a microreaction vessel. The mixture was heated at 120 °C for 10 min. The crude product was then diluted with water and purified by solid-phase extraction (SPE) using a silica-based C18 cartridge (Waters Corporation, Milford, MA, USA). The radiolabeled product was eluted with 1.5 mL of 50% ethanol, followed by 1 mL of saline. The radiochemical yield exceeded 90%, and the final radiochemical purity was greater than 95%. For the quantification of radioligand uptake, LNCaP and VCap were seeded on 24-well plates 24 h prior to treatment. Cells were then incubated with 0.1 MBq of ^177^Lu-PSMA-617 for 4 h at 37 °C. After incubation, the solution was removed and cells were cultivated in the corresponding media for 20 h. Subsequently, the uptake was determined by using a gamma counter (Wizard2) (PerkinElmer, Massachusetts, MA, USA). To calculate the effectively applied dose (AD), the number of viable cells remaining after incubation was determined and correlated with the corresponding gamma counter values. Gamma counter tubes (Sarstedt, Nümbrecht, Germany) were used for all measurements and respective controls were included.

### 4.4. Proliferation Assay (MTS), Live Dead Staining, and Cell Cycle Analysis

Cell proliferation was characterized after treatment using the MTS colorimetric assay kit (Abcam, Cambridge, UK, ab197010). For this purpose, cells were seeded in 96-well plates 24 h prior to treatment. After treatment, 20 µL of MTS reagent was added to each well and incubated at 37 °C for 30 min. Absorbances were measured using a SpectraMax^®^ iD3 Microplate Reader (Molecular Devices, San Jose, CA, USA, SoftMax Pro7.1.2) at 460 nm. Relative cell proliferations were determined against appropriate controls.

Cell viability of VCaP cells was additionally quantified by the LIVE/DEAD™ Viability/Cytotoxicity Kit (Thermo, L3224). For this, cell viability staining was performed according to the manufacturer’s protocol. Briefly, cells were washed with 500 µL 1× PBS, trypsinized, and centrifuged. Supernatants were discarded, and cells were carefully resuspended in 250 µL of medium. Subsequently, 0.25 µL of the reactive dye was added. Incubation was performed for 30 min on ice, protected from light. Samples were washed twice with 500 µL 1× PBS via centrifugation. For determination of cell viability, an untreated control and a positive dead control (treated with 500 µL of 70% ethanol for 2 min at RT) were included. Flow cytometric analysis was performed in 500 µL 1× PBS using the FACSCanto^TM^ Clinical Flow Cytometry System II (BD Biosciences, San Jose, CA, USA). Cell populations were gated separately by cell granularity and size (forward (FSC) and side scatter (SSC)) against corresponding controls. At least 10,000 events were analyzed using appropriate flow cytometer instrument settings.

Assessing the impact of GE on the cell cycle, PCa cells were seeded on 24-well plates 24 h prior to GE treatment. Treatment was performed with GE2 for 2 h with respective controls included. Following the incubation, one drop of NucBlue™ Live ReadyProbes™ Reagent (Hoechst 33342) (Thermo, R37605) was added to the cells and incubated for 10 min at 37 °C. Cells were harvested by trypsinization and washed with 1× PBS through centrifugation.

Flow cytometry was performed for viability stain and cell cycle analysis in 300 µL 1× PBS. Detached cells in supernatants were included in both analysis through centrifugation. Cell viability analysis and DNA amount quantification was carried out using the FACSCanto™ Clinical Flow Cytometry System II (BD Biosciences). Forward scatter (FSC) and side scatter (SSC) were used to gate each cell line separately, excluding debris and aggregates. Fluorescence intensities were measured using the integrated 3-laser, 8-color (2-4-2) laser module, with appropriate instrument settings for at least 10,000 cell events. Data were analyzed using FACSCanto™ Clinical Software (Version v9.0.1). For determination of cell viability, respective controls and a positive dead control (treated with 500 µL of 70% ethanol for 2 min at RT) were included. Cell populations were calculated relatively against respective controls for each cell line. For cell cycle analysis, fluorescence intensity was plotted separately for each cell line in a histogram to determine the distribution (%) of cells across the G0/G1, S, and G2/M cell cycle phase.

### 4.5. Comet Assay (DNA Damage) Analysis

The number of DNA-damaged cells was quantified by alkaline Comet assay. Cells were seeded 24 h prior to treatment. Post treatment, cells were detached from substrates and after centrifugation (see section cell culture), resuspended in 30 µL of media. For each sample, 20 µL of cell suspension was carefully mixed with 180 µL of 0.5% low-melting point agarose (*w*/*v*) (LMP-A) (Sigma, Darmstadt, Germany A9045) diluted in 1× PBS. Next, 60 µL of this mixture was added onto 1.5% LMP-A (*w*/*v*) diluted in 1× PBS pre-coated microscope slides, mounted with a coverslip and dried for 5 min at 4 °C. Coverslips were removed and cells were lysed in lysis buffer (2.5 M NaCl, 0.1 M EDTA, 10 mM Tris, pH 10 in deionized water) for 1 h in the dark at 4 °C. Unwinding of DNA was performed in alkaline solution (0.3 M NaOH, 1 mM EDTA, pH 13 in deionized water) for 20 min in the dark at 4 °C, followed by electrophoresis for 20 min at 25 V. Slides were neutralized for 5 min in wash buffer (0.4 M Tris diluted in deionized water) and afterwards, washed with deionized water. Then, 50 µL of diluted GelRed^TM^ (1:400) (*v*/*v*) (VWR, 41001.) in staining buffer (1 deionized water/3 DABCO (Roth, Karlsruhe, Germany 0718.1) (*v*/*v*)) was added. Samples were mounted with a coverslip and stored in the dark at 4 °C until microscopy.

### 4.6. Immunefluorescence Staining

For immunefluorescence, staining cells were seeded 48 h prior to treatment on glass bottom substrates (Ibidi GmbH, München, Germany). After treatment, cells were fixed with 4% formaldehyde (Otto Fischar, Schoenaich, Germany, 02653048) for 30 min at 37 °C and incubated with 30 mM glycine (*v*/*v*) (Sigma, 67419) diluted in 1× PBS for 10 min at RT. Membranes were permeabilized with 2% Triton X-100 (*v*/*v*) (Merck, X100) diluted in 1× PBS for 3 min at RT. Afterwards, cells were washed with 1× PBS for 5 min. Non-specific protein binding was blocked with 5% (*w*/*v*) bovine serum albumin (BSA) (Merck, A-7906) diluted in 1× PBS (blocking buffer) for 45 min at RT. Primary antibodies Anti-Integrin β1 (1:200) (Abcam, EP1041Y), Anti-androgen receptor (1:150) (Merck, 06-680), 488/530 nm FAM^®^-conjugated LETD-FMK Caspase-8 reagent (1:150) (Vybrant^TM^ FAM^TM^ Caspase-8 Assay kit) (Thermo, V35119) were incubated in blocking buffer over night at 4 °C. Cells were washed three times with blocking buffer for 5 min at RT. Subsequently secondary antibody (Goati anti-Mouse IgG (H+L) Alexa Fluor^TM^ 568 nm (1:400) (Thermo, A-11004), Goat anti-Rabbit IgG (H+L) Alexa Fluor^TM^ 488 nm (1:400) (Thermo, A-11008) was diluted in blocking buffer and added to cells for 1 h at RT. Again, cells were washed. Nuclei staining was performed by incubation of Hoechst 33342 Ready Flow Reagent (Thermo, R37165) diluted 1:20,000 (*v*/*v*) in 1× PBS for 5 min at RT, followed by washing steps. Stained cells were sealed with coverslips (Fisher Scientific, Hampton, NH, USA 15767572) and Immu-Mount (Epredia, Hempel Hemstead, UK, 9990402) for preservation until microscopy. All incubation steps were performed on a 2D shaker at 20 rpm.

### 4.7. Microscopy and Image Analysis

All microscope and image analysis settings were adjusted accordingly to the dyes used and kept identical within each specific experiment to ensure maximum comparability. A Leica DM IL LED microscope (Leica-Microsystems, Wetzlar, Germany) equipped with a pE-300lite light source, a HIN PLAN CY 10×/0.25 objective, a K3M camera, and LAS X-software (version 3.7.4.23463) was used for fluorescence imaging to analyze DNA damage after treatment. At least 50 cells of each sample were imaged, and Comets were quantified using a Comet assay analysis software (CometAssay^TM^ Trevigon, Version 1.3d, R&D System, Minneapolis, MN, USA). The Axio Observer microscope (Carl Zeiss AG, Oberkochen, Germany) with a Colibri 5 light source, a Plan Apochromat 40×/1.4 oil objective, an Axiocam 503 mono, an Apotome 3, and Zen 3.8 software was utilized for multi-layer imaging (z-stack = 10 layers) of immunefluorescence samples. Images were processed and analyzed with ImageJ software (version v1.54f). Quantification of protein expressions by total fluorescence intensity (AR), characterization of protein localization profiles (Integrin β1), as well as protein colocalizations (Integrin β1 and Caspase-8) were analyzed as described by Hoffmann et al. [56].

### 4.8. Characterization of Protein Alterations

#### 4.8.1. Protein Isolation

All protein isolation steps were performed on ice. Cells were seeded 48 h before isolation. Cells were mechanically detached into 1 mL ice-cold 1× PBS using a cell scraper. Cells were centrifuged at 100× *g* and 4 °C for 5 min; the supernatant was discarded. After washing cells with 0.5 mL ice-cold 1× PBS and another centrifugation, 150 µL of RIPA-lysis buffer supplemented with protease and phosphatase inhibitor was added. A lysis buffer with inhibitors was prepared as described previously [56]. Subsequently, cells were vortexed, homogenized with a syringe (inner diameter = 0.26 mm), and lysed on ice for 30 min. Lysates were centrifuged at 21,300× *g* and 4 °C for 15 min. Supernatants (protein lysates) were stored at −80 °C until use.

#### 4.8.2. Determination of Protein Concentrations

Total protein concentrations were determined using the Micro BCA^TM^ Protein-Assay-Kit (Thermo, 23235) accordingly to the manufacture’s protocol. Samples were diluted 1:30 in RIPA-lysis buffer and mixed with the kit included Working Reagent (WR) before measurement. Absorbances were measured with a SpectraMax^®^ iD3 Microplate Reader at 562 nm. Protein concentrations were determined using a BCA-dilution series-standard curve.

#### 4.8.3. SDS-PAGE

Prior to gel electrophoresis, protein lysates were diluted with RIPA-lysis buffer to identical protein concentrations. Subsequently, protein lysates were diluted in a total volume of 20 µL loading dye (Bio-Rad Laboratories, Hercules, CA, USA, 1610747) (3 sample/1 loading dye (*v*/*v*)) supplemented with β-mercaptoethanol (Merck, 8.05740.0250) (1 β-mercaptoethanol/9 loading dye (*v*/*v*)). Protein denaturation was performed for 5 min at 72 °C. Samples were loaded onto a TGX stain-free protein gel (Bio-Rad, 4568026). As a protein standard, 3 µL of Precision Protein All Blue Prestained Protein Standards (Bio-Rad, 1610373) was used. Protein separation was performed at 120 V for 2 h.

#### 4.8.4. Western Blot and Antibody Stain

For protein blotting, an LF-PVDF membrane and transfer stacks (Bio-Rad, 1704274) were used. Preparation of the blotting procedure was performed as given by the manufacturer’s protocols. Briefly, the blot was placed in a Trans-Turbo Transfer System (BioRad, 1704150) for 7 min at 1.3 A and 25 V using the preinstalled mixed-molecular weight (MW) program (Turbo Transfer, MIXED MW). Unspecific protein binding sites were saturated by using a blocking buffer consisting of 5% (*w*/*v*) BSA (Merck, A-7906) and 0.05% (*v*/*v*) Tween 20 (Merck, P1379) diluted in 1× PBS. The blocking buffer was incubated over night at 4 °C. All incubation steps were carried out on a 2D shaker with 20 rpm.

Primary Bcl-2 antibody (antibodies online, ABIN2857047) was diluted 1:400 (*v*/*v*) in blocking buffer and incubated over night at 4 °C. After washing three times with blocking buffer for 5 min at room temperature (RT), the secondary Goat Anti-Rabbit IgG StarBright Blue 700 (Bio-Rad, 12004161) was added in a dilution 1:400 (*v*/*v*) in blocking buffer to membranes. Incubation was performed for 1 h at RT. Again, membranes were washed with blocking buffer. Imaging was performed at an emission wavelength of 700 nm using the ChemiDoc MP Imaging System (Bio-Rad, 12003154). Stain-free protein gels were used to implement the total amount of protein transferred to LF-PVDF membranes. Quantification of detected Bcl-2 protein levels after treatment was carried out against untreated controls with ImageLab software (Bio-Rad, Version 6.1).

### 4.9. Characterization of mRNA Alterations

#### 4.9.1. RNA Isolation

For RNA isolation from cell cultures, the RNeasy-Midi Kit (Qiagen, Hilden, Germany, 77144) was used. Cells were cultured for 24 h prior to treatment. Afterwards, RNA isolation was processed according to the manufacturer’s instructions. RNA concentrations were measured using a NanoDrop Lite Plus System (Thermo, NDLPLUSGL) at 260 nm. Isolated RNA was stored at −80 °C until cDNA synthesis.

#### 4.9.2. cDNA Synthesis

For the cDNA synthesis of isolated RNA, the Maxima First Strand cDNA-Synthesis Kit (Thermo, K1641) was used. All preparation steps were performed on ice according to the given standard protocol. Briefly, 300 ng of isolated RNA was diluted with nuclease-free water to a final volume of 14 µL. Afterwards, 4 µL of 5× Reaction Mix and 2 µL of Maxima Enzyme Mix were added. Finally, cDNA synthesis was performed in a thermocycler (Biometra TRIO, Jena, Germany) by incubation at 25 °C for 10 min, followed by 50 °C for 30 min and 85 °C for 5 min. Subsequently, reaction mixes were cooled down to 4 °C. Synthesized cDNAs were diluted with 20 µL of nuclease-free water and stored at 4 °C until qRT-PCR analysis.

#### 4.9.3. qRT-PCR

For qRT-PCR experiments, TaqMan^®^ Assay against AR (Thermo, Hs00171172_m1) and CYP2C9 (Thermo, Hs02383631_s1) were used. Glyceraldehyde 3-phosphate dehydrogenase (GAPDH) (Thermo, Hs02786624_g1) served as internal standard. For each sample, a Mastermix consisting of 10 µL TaqMan^®^ Advanced Master-Mix (Thermo, 4444964), 1 µL of TaqMan^®^ Assay, and 6 µL of nuclease-free water were added to a 96-well plate. Finally, 3 µL of synthesized cDNA was applied. DNA replication was performed using the StepOnePlus Real Time PCR System (Thermo, 4376600) by heating the mixtures to 95 °C for 20 s, followed by 40 cycles of cooling to 60 °C for 20 s and heating up to 95 °C for 1 s. The qRT-PCR system-integrated StepOne Software (version 2.3) was used for evaluation.

### 4.10. CYP2C9 Activity Assay

To quantify metabolic activity after treatment, the fluorometric CYP2C9 Activity assay Kit (Abcam, ab211072) was used. Kit reagents were prepared as described in the manufacturer’s protocol. Cells were cultured in 24-well plates for 24 h before treatment. After treatment, cells were detached by incubation with trypsin-EDTA solution as described (see section cell culture). Supernatants were discarded and cell pellets were resuspended in 500 µL of ice-cold Assay Buffer and incubated at 4 °C for 5 min. Samples were centrifuged at 15,000× *g* and 4 °C for 15 min. Subsequently, supernatants were collected (48 µL) and mixed with 2 µL of 100× NADPH Generating System in a 96-well plate. Those sample mixtures were then incubated at 37 °C for 30 min. During this incubation, the Reaction Assay Solution consisting of 5 mM CYP2C9 stock solution (6 µL), 100× β-NADP^+^ stock solution (50 µL), and Assay Buffer (1444 µL) was freshly prepared. Then, 30 µL of this mixture was added to samples. To ensure dynamic recording, absorbances were directly measured every 5 min for 1 h at 37 °C using a SpectraMax^®^ iD3 microplate reader at 502 nm. Finally, CYP2C9 Activity quantification was performed as described in the manufacturer’s protocol.

### 4.11. Calculation of Combination Indices (CIs)

The combination indices were calculated using the formula:$$CI=\frac{E_{A}+E_{B}}{E_{AB}}$$

In this context, *E* represents the magnitude of the effect exerted on the sample. *E_A_* corresponds to the effect of substance *A*, while *E_B_* notes the effect of substance *B*. *E_AB_* describes the effect observed from the combination treatment.

Traditionally, *CI* values greater than 1 are considered antagonistic [57]. However, to achieve a more refined discrimination among potentially antagonistic substances, we implemented the following thresholds: *CI* values ≤ 0.8 indicate synergism, values between 0.8 and 1.6 indicate additivity, and values ≥ 1.6 indicate antagonism. This adjustment allows us to better resolve subtle differences between therapeutic agents and to identify promising candidates for further combination testing in preclinical models.

### 4.12. Statistics and Graphical Analysis

All data are given as mean values with standard deviations from at least three independent experiments. Relative calculated data were generated from each experiment against corresponding controls and combined accordingly. Mean values and standard deviations were generated with Microsoft Excel (MS Office 2019). Statistical analysis was performed using a one-way analysis of variance (ANOVA) with Microsoft Excel. A *p*-value of <0.05 was considered as significant. Only significances of <0.05, <0.01, and <0.001 were marked with one to three asterisks. Figures and graphs were created using Microsoft PowerPoint (MS Office 2019) and Origin 2019 64Bit (OriginLab Graphing & Analysis, Northampton, MA, USA).

## Figures and Tables

**Figure 1 ijms-26-01777-f001:**
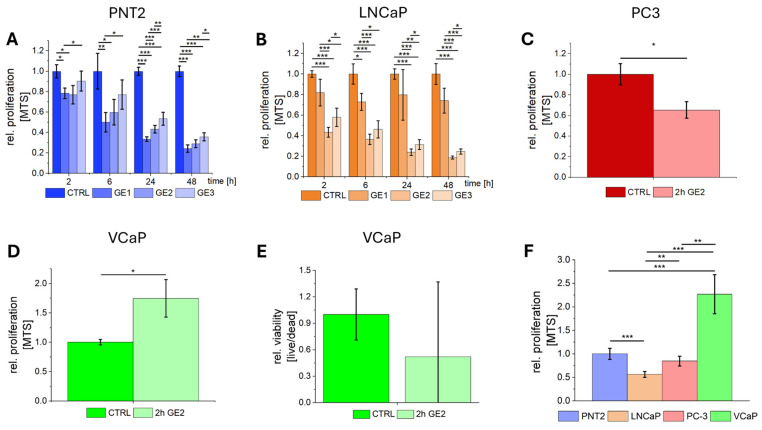
Influence of GE on cell proliferation. The relative proliferation after GE treatment over 48 h was analyzed for GE concentrations GE1, GE2, and GE3 using MTS for PNT2 (**A**) and LNCaP (**B**). The relative proliferation of PC3 (**C**) and VCaP (**D**) after 2 h of GE2 treatment are compared to untreated controls. Additionally, live/dead-staining was performed for VCaP cells to quantify viability after GE2 treatment (**E**). Selectivity of GE for PCa cells is characterized by the relative proliferation of LNCaP, PC3, and VCaP compared to the healthy cell line PNT2 (**F**). The statistical significances are marked by asterisks (*: *p* < 0.05; **: *p* < 0.01; ***: *p* < 0.001).

**Figure 2 ijms-26-01777-f002:**
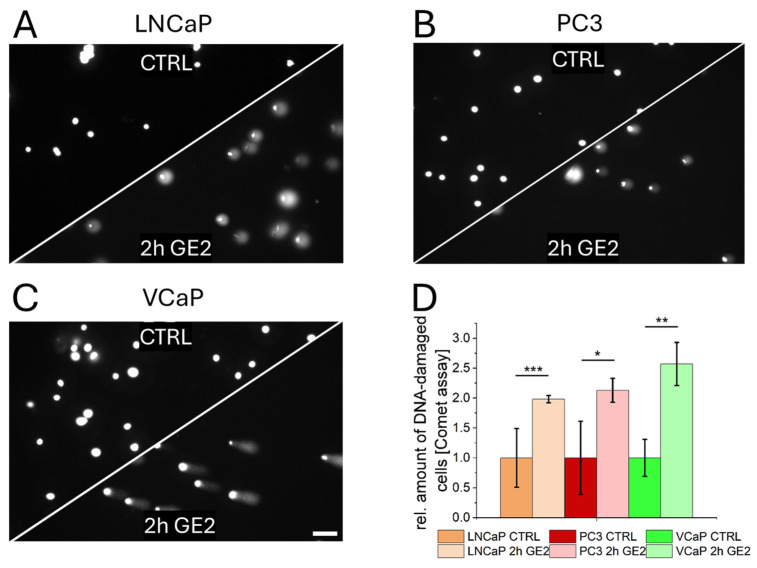
Changes in DNA integrity by GE treatment. Fluorescence microscopy of the Comet assay of LNCaP (**A**), PC3 (**B**), and VCaP (**C**) cells without GE (CTRL) and with GE2 treatment for 2 h (2 h GE2) (**D**). Percentage of DNA-damaged cells from controls (CTRL) and treated (2 h GE2) samples in LNCaP, PC3, and VCaP (**D**). The statistical significances are indicated by asterisks (*: *p* < 0.05; **: *p* < 0.01; ***: *p* < 0.001). Scalebar = 100 µm.

**Figure 3 ijms-26-01777-f003:**
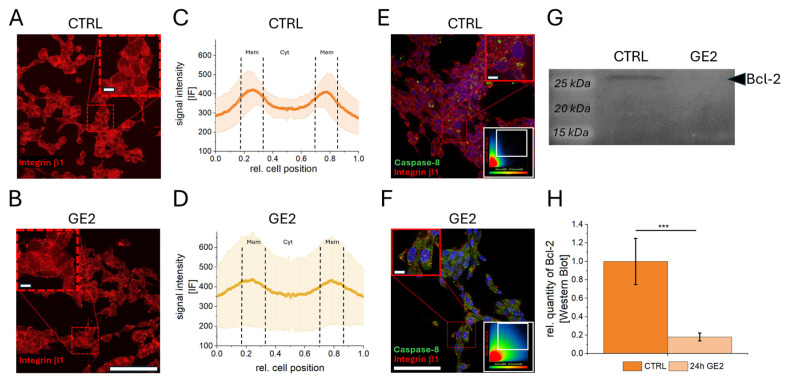
Changes of Integrin β1 and Caspase-8 expression and localization. Fluorescence microscopy of Integrin β1 immunefluorescence (IF) in control samples ((CTRL; (**A**)) and 24 h GE-treated LNCaP cells (**B**) with zoom in images (red boxes). Cellular profiles indicate signal localization within cellular plasma membrane (Mem) and cytoplasm (Cyt) (**C**,**D**). Overlay images of Integrin β1 (red) and Caspase-8 (green) with zoom in images (red boxes) and colocalization plots (white boxes) (**E**,**F**). The Western blot image (**G**) indicates protein bands of anti-proliferative protein Bcl-2 (28 kDa) and a relative quantification in (**H**). The statistical significances are indicated by asterisks (***: *p* < 0.001). Scalebar = 100 µm/zoom in images 10 µm.

**Figure 4 ijms-26-01777-f004:**
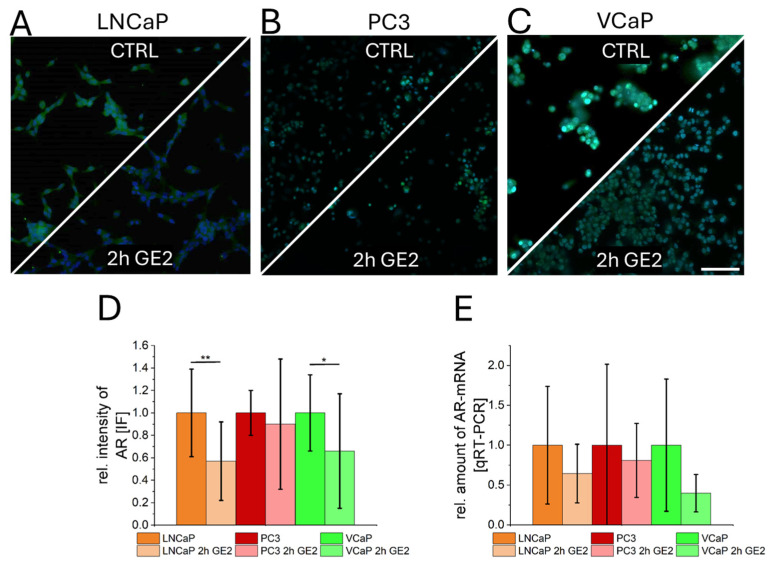
Influence of GE-treatment on PCa-Androgen receptors (AR). Immunefluorescence of AR ((**A**–**C**), green) and fluorescence intensity quantification (**D**) indicate reduced AR-membrane localization after GE2-treatment for LNCaP and VCaP. The AR-negative cell line PC3 shows no significant differences. A quantification of AR mRNA amount by qRT-PCR showed no significant differences for all cell types after GE treatment (**E**). The statistical significances are indicated by asterisks (*: *p* < 0.05; **: *p* < 0.01). Scalebar = 100 µm.

**Figure 5 ijms-26-01777-f005:**
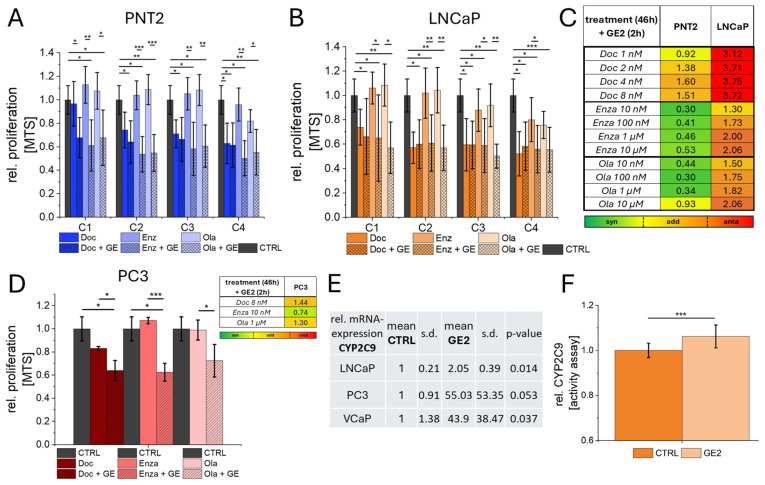
Combination therapy of GE2 and established therapeutics against PCa. Increasing concentrations (C1–C4) of the therapeutics Docetaxel (Doc—chemotherapeutic agent), Enzalutamide (Enza—ADT), and Olaparib (Ola—PARPi) were investigated by MTS assay regarding their anti-proliferative and combined effects with GE (+GE) for healthy prostate epithelial cells (PNT2; (**A**)) and PCa cells (LNCaP; (**B**)). Cells were treated with increasing concentrations of therapeutic agents (C1–C4) for 46 h, and in combination with GE, the incubation of GE2 was carried out for 2 h (treatment (46 h) + GE2 (2)). The resulting quantification of effects is shown in (**C**), where synergistic effects were assigned up to a value of 0.8, additive effects at values from 0.8 to 1.6, and antagonistic effects for values higher than 1.6. The analysis for PC3 cells was performed using the combination of the most effective Doc, Enza, and Ola concentrations with GE (**D**). The characterization of CYP2C9 with qRT-PCR (**E**) and activity assay (**F**) was used to investigate the reduced efficacy of the therapeutic agents under GE administration in PCa cells. The statistical significances are indicated by asterisks (*: *p* < 0.05; **: *p* < 0.01; ***: *p* < 0.001). Scalebar = 100 µm.

**Table 1 ijms-26-01777-t001:** Cell culture media and supplements for all cell lines. Amounts are given as (*v*/*v*).

Cell Line	Media	Supplemented with
LNCaP	RPMI 1640 ^(1)^	1% of 10,000 U/mL penicillin-streptomycin (PS) ^(2)^ 10% of fetal bovine serum (FBS) ^(3)^
PC3	DMEM ^(1)^	
VCaP	DMEM F12 ^(1)^	
PNT2	RPMI 1640 ^(1)^	1% PS 10% FBS 1% of 200 mM L-glutamine ^(2)^

Purchased from PanBiotech ^(1)^, Thermo ^(2)^, or Merck ^(3)^.

## Data Availability

The data that support the findings of this study are available from the corresponding authors upon reasonable request.

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
