# Peer review of "Mechanism of Action and Interaction of Garlic Extract and Established Therapeutics in Prostate Cancer"

_ijms, 2025, doi:10.3390/ijms26041777_

Round 1
Reviewer 1 Report
Comments and Suggestions for Authors
The article titled “Mechanism of action and interaction of garlic extract and established therapeutics in prostate cancer” investigates the effects of garlic extract (GE) on prostate cancer (PCa) cells and its interactions with established PCa treatments. Inspired by a case report of a patient who maintained low PSA levels with daily GE intake, the authors conducted in vitro experiments using healthy prostate epithelial cells (PNT2) and PCa cell lines (LNCaP, PC3, VCaP). They found that GE reduced proliferation in LNCaP and PC3 cells compared to PNT2 cells, but increased proliferation in VCaP cells. GE induced DNA damage in all PCa cell lines and altered the expression and localization of key proteins such as Integrin β1, Caspase-8, and Bcl-2. The study revealed potential antagonistic effects when GE was combined with established PCa therapies, including chemotherapy, androgen deprivation therapy (ADT), and PARP inhibitors (PARPi). This antagonism was attributed to GE-induced upregulation of the metabolic enzyme CYP2C9 in PCa cell lines, which may affect drug metabolism. However, the research could be improved by including in vivo studies to corroborate the in vitro results and provide a more comprehensive understanding of GE's effects in a physiological context. The authors emphasize the need for careful assessment of natural extracts in cancer treatment, especially when combined with established therapies, highlighting the importance of considering potential herb-drug interactions in cancer care. The reviewer has the following comments that authors need to address.
1. The authors have provided an overview of allicin-related studies in the introduction; however, a more detailed discussion on the active components of garlic extract responsible for its anticancer properties would strengthen the study. Additionally, it would be beneficial to clarify whether any characterization studies were conducted to identify these active constituents. Including this information would enhance the impact and scientific depth of the manuscript.
2. To provide a more comprehensive understanding of Garlic Extract's (GE) impact on drug metabolism, the investigation of CYP450 enzymes should extend beyond CYP2C9 in future studies. Exploring the effects on other key CYP isoforms, such as CYP3A4, CYP2D6, and CYP1A2, would offer a more complete picture of potential herb-drug interactions and metabolic implications.
3. To strengthen the study, consider mentioning the dose-response studies for the combination treatments to determine the optimal ratios of Garlic Extract (GE) and conventional therapies. This would provide valuable insights into potential synergistic effects, dose optimization, and therapeutic efficacy, ultimately enhancing the clinical relevance of the findings.
4. To provide a more comprehensive assessment of potential interactions, the authors are encouraged to include additional prostate cancer (PCa) therapeutics in the combination studies. Incorporating PSMA-targeting therapeutics, as reported in the following recent articles, would offer valuable insights into the broader applicability and therapeutic potential of Garlic Extract (GE) in PCa treatment.
https://www.cell.com/molecular-therapy-family/nucleic-acids/fulltext/S21622531(24)00080-5
https://www.mdpi.com/1422-0067/25/4/2123
5. To further enhance the study, the authors are encouraged to investigate the molecular mechanisms underlying the differential response of VCaP cells to Garlic Extract (GE) treatment compared to other prostate cancer (PCa) cell lines. Understanding these mechanistic differences could provide valuable insights into the specific pathways influenced by GE and its potential implications for targeted PCa therapy.
6. The reviewer suggests improving the quality of the figures 1,3 and 5 with a consistent font to enhance clarity and visual impact.
Author Response
Reviewer 1:
The article titled “Mechanism of action and interaction of garlic extract and established therapeutics in prostate cancer” investigates the effects of garlic extract (GE) on prostate cancer (PCa) cells and its interactions with established PCa treatments. Inspired by a case report of a patient who maintained low PSA levels with daily GE intake, the authors conducted in vitro experiments using healthy prostate epithelial cells (PNT2) and PCa cell lines (LNCaP, PC3, VCaP). They found that GE reduced proliferation in LNCaP and PC3 cells compared to PNT2 cells, but increased proliferation in VCaP cells. GE induced DNA damage in all PCa cell lines and altered the expression and localization of key proteins such as Integrin β1, Caspase-8, and Bcl-2. The study revealed potential antagonistic effects when GE was combined with established PCa therapies, including chemotherapy, androgen deprivation therapy (ADT), and PARP inhibitors (PARPi). This antagonism was attributed to GE-induced upregulation of the metabolic enzyme CYP2C9 in PCa cell lines, which may affect drug metabolism. However, the research could be improved by including in vivo studies to corroborate the in vitro results and provide a more comprehensive understanding of GE's effects in a physiological context. The authors emphasize the need for careful assessment of natural extracts in cancer treatment, especially when combined with established therapies, highlighting the importance of considering potential herb-drug interactions in cancer care. The reviewer has the following comments that authors need to address.
Comment 1: The authors have provided an overview of allicin-related studies in the introduction; however, a more detailed discussion on the active components of garlic extract responsible for its anticancer properties would strengthen the study. Additionally, it would be beneficial to clarify whether any characterization studies were conducted to identify these active constituents. Including this information would enhance the impact and scientific depth of the manuscript.
Response 1: We sincerely appreciate this reviewer´s insightful comment and the opportunity to refine our manuscript accordingly. You are right that we did not explicitly discuss the specific constituents of GE responsible for the various effects observed in our in vitro studies. Nevertheless, we fully acknowledge your point that a more detailed introduction of the already characterized and well-known components of GE would aid readers in better understanding the outcomes of our analysis. Due to the numerous interactions and diverse constituents of the extract, we refrained from conducting a differential analysis of the patient-prepared extract. However, given the comparable preparation method to the referenced publication [55], we can reasonably assume an identical composition of constituents.
To address this, we have added a dedicated paragraph (line 47-63) to the introduction that highlights the key constituents of GE and their previously described effects on tumor cells. This addition aims to provide a clearer context for interpreting and linking our experimental findings to the components of GE.
Comment 2: To provide a more comprehensive understanding of Garlic Extract's (GE) impact on drug metabolism, the investigation of CYP450 enzymes should extend beyond CYP2C9 in future studies. Exploring the effects on other key CYP isoforms, such asCYP3A4, CYP2D6, and CYP1A2, would offer a more complete picture of potential herb-drug interactions and metabolic implications.
Response 2: We fully agree with your point on this matter as well. However, we deliberately focused on CYP2C9, as this variant plays the most significant role in interactions with therapeutics and has already been well described in the literature [see Ref. 31].
Nevertheless, we have expanded the discussion to acknowledge this limitation (see line 271-273).
Comment 3: To strengthen the study, consider mentioning the dose-response studies for the combination treatments to determine the optimal ratios of Garlic Extract (GE) and conventional therapies. This would provide valuable insights into potential synergistic effects, dose optimization, and therapeutic efficacy, ultimately enhancing the clinical relevance of the findings.
Response 3: Thank you very much for this valuable comment. We have aimed to align our experimental design with the described case report. Specifically, we first aimed to determine the concentration of GE that effectively influences the used PCa cell lines. This defined concentration was then used in our subsequent in vitro studies. The concentration GE2 represents the minimally necessary and effective dose that elicited measurable effects on the used PCa cell lines.
In the next step, we combined this fixed concentration with a concentration series of the different established therapeutics to treat PCa. We fully agree with your point that extrapolating these in vitro doses to clinically relevant concentrations remains speculative. In this regard, animal studies could provide further insights into key pharmacokinetic and pharmacodynamic parameters. The exact concentrations that tumor cells might reach in vivo and the corresponding required daily dose are highly variable and depend on multiple factors (e.g. route of administration).
We appreciate your valuable feedback, which has helped us to refine the discussion on this aspect (see line 285-294).
Comment 4: To provide a more comprehensive assessment of potential interactions, the authors are encouraged to include additional prostate cancer (PCa) therapeutics in the combination studies. Incorporating PSMA-targeting therapeutics, as reported in the following recent articles, would offer valuable insights into the broader applicability and therapeutic potential of Garlic Extract (GE) in PCa treatment.
https://www.cell.com/molecular-therapy-family/nucleic-acids/fulltext/S21622531(24)00080-5
https://www.mdpi.com/1422-0067/25/4/2123
Response 4: Thank you very much for this exciting idea. Unfortunately, we do not have access to these highly innovative and intriguing therapeutic approaches. Nevertheless, we have tried to incorporate the fundamental principles of receptor targeted therapies by using the FDA-approved 177Lu-PSMA-617 radioligand therapy in combination with GE, as this important therapeutic strategy should, of course, also be considered in this context.
Accordingly, we have included the relevant results in the supplementary material (Suppl. Figure S5). The mentioned innovative miRNA-therapies, which also could use PSMA as target receptor for miRNA-delivery, have been referenced in the discussion section (line 279-282).
We truly appreciate your insightful suggestions, which have helped to further enhance the translational acceptance of our manuscript.
Comment 5: To further enhance the study, the authors are encouraged to investigate the molecular mechanisms underlying the differential response of VCaP cells to Garlic Extract (GE) treatment compared to other prostate cancer (PCa) cell lines. Understanding these mechanistic differences could provide valuable insights into the specific pathways influenced by GE and its potential implications for targeted PCa therapy.
Response 5: VCaP cells originate from vertebral metastasis and are therefore among the most aggressive cell lines of PCa. In other studies, we have also validated their low sensitivity towards approaches such as Enzalutamide and PSMA radioligand therapy. To characterize the results presented here and to understand the underlying mechanisms, we compared the cell cycle of VCaP cells to other PCa lines LNCaP and PC3 after GE treatment by quantification cell cycle phases using nuclei staining and flow cytometric analysis. Cell cycle analysis revealed that, despite DNA damage, VCaP cells remain in the G1 phase after GE-treatment at higher levels compared to all other PCa cell lines tested in this study (Suppl. Figure S2). We would like to thank you for your helpful comment, which allowed us to complement these important data in the manuscript.
Comment 6: The reviewer suggests improving the quality of the figures 1,3 and 5 with a consistent font to enhance clarity and visual impact.
Response 6: Thank you for your attentive review of our manuscript. We have adjusted the style of the corresponding figures accordingly and transferred the style as reference to all other figures. We hope that our figures are now consistent and meet your requirements.
Reviewer 2 Report
Comments and Suggestions for Authors
This paper explores the interaction between garlic extract (GE) and prostate cancer (PCa) treatment, providing valuable in vitro experimental data, particularly in terms of the innovative findings regarding the impact of GE on CYP2C9 enzyme expression and its potential drug interactions. However, the paper has the following critical issues that require major revisions to meet publication standards:
1. Preparation method of garlic extract (GE) lacks key parameters: In Line 272, the preparation method of GE does not specify critical parameters (e.g., solvent,), which affects the reproducibility of the experiment. It is necessary to supplement the chemical composition analysis of GE (e.g., HPLC detection of allicin content), clarify the actual concentrations of active ingredients corresponding to different concentrations (GE1-GE3), and compare them with existing studies.
2. Insufficient explanation for the contradictory mechanism of VCaP cell proliferation: VCaP cells show increased proliferation after GE treatment, but DNA damage (comet assay) is also significantly increased, resulting in contradictory findings. Further exploration of potential mechanisms (e.g., cellular senescence, activation of pro-survival signaling pathways) is needed, or cell cycle analysis (flow cytometry) should be performed to verify whether G1 phase arrest occurs.
3. Causal relationship between CYP2C9 upregulation and drug efficacy is not validated: The upregulation of CYP2C9 is speculated to be the reason for the reduced efficacy of chemotherapy drugs, but direct evidence is lacking (e.g., CYP2C9 inhibitor or gene silencing experiments).
4. Insufficient discussion on the limitations of clinical cases: A single case report (PSA level reduction) serves as the starting point of the study, but its small sample size and potential for significant error are not adequately addressed. It is necessary to clarify the preliminary observational nature of this case or supplement the discussion with more clinical research data to support the conclusions.
5. Incomplete statistical methods: The paper does not mention whether multiple comparison corrections (e.g., Tukey HSD) were performed, which may increase the risk of false positives. It is recommended to provide details on the statistical methods, specify the correction methods used, and annotate specific p-values in the figures and tables.
6. Optimization needed for figures and terminology: In Figure 5, the scoring criteria for "synergistic/antagonistic effects" (thresholds of 0.8 and 1.6) are not clearly justified; some figures (e.g., immunofluorescence images) have insufficient resolution. It is recommended to provide high-resolution images and ensure clarity in all visual data.
Author Response
Reviewer 2
This paper explores the interaction between garlic extract (GE) and prostate cancer (PCa) treatment, providing valuable in vitro experimental data, particularly in terms of the innovative findings regarding the impact of GE on CYP2C9 enzyme expression and its potential drug interactions. However, the paper has the following critical issues that require major revisions to meet publication standards:
Comment 1: Preparation method of garlic extract (GE) lacks key parameters: In Line 272, the preparation method of GE does not specify critical parameters (e.g., solvent,), which affects the reproducibility of the experiment. It is necessary to supplement the chemical composition analysis of GE (e.g., HPLC detection of allicin content), clarify the actual concentrations of active ingredients corresponding to different concentrations (GE1-GE3), and compare them with existing studies.
Response 1: We fully agree with the reviewer´s suggestion that the preparation method should be described more clearly. Accordingly, we have added relevant details to the Materials and Methods section (line 324-327).
Due to the numerous interactions and diverse constituents of the extract, we refrained from conducting a differential analysis of the patient-prepared extract. However, for the GE used in our in vitro studies, the given preparation method to the referenced publication [55], we can reasonably assume an identical composition of constituents as described previously.
Since GE1-3 represents dilutions of a common stock solution, a corresponding serial concentration dependent effect can also be assumed.
We very much appreciate this valuable feedback, which has helped improve the clarity and rigor of our manuscript and added a dedicated paragraph (line 47-63) to the introduction.
Comment 2: Insufficient explanation for the contradictory mechanism of VCaP cell proliferation: VCaP cells show increased proliferation after GE treatment, but DNA damage (comet assay) is also significantly increased, resulting in contradictory findings. Further exploration of potential mechanisms (e.g., cellular senescence, activation of pro-survival signaling pathways) is needed, or cell cycle analysis (flow cytometry) should be performed to verify whether G1 phase arrest occurs.
Response 2: We fully agree with this important comment and sincerely appreciate the valuable feedback. VCaP cells exhibit varying responses to therapeutics in nearly all of our analysis. Understanding this phenomenon in greater depth remains a key focus for our future investigations.
To generate an initial insight, we have now conducted comparative nucleic staining to quantify cell cycle arrest in LNCaP, PC3 and VCaP cells. Cell cycle analysis revealed that, despite DNA damage, VCaP cells remain in the G1 phase after GE-treatment at higher levels compared to all other cell types tested in this study (Suppl. Figure S2).
Comment 3: Causal relationship between CYP2C9 upregulation and drug efficacy is not validated: The upregulation of CYP2C9 is speculated to be the reason for the reduced efficacy of chemotherapy drugs, but direct evidence is lacking (e.g., CYP2C9 inhibitor or gene silencing experiments).
Response 3: We acknowledge that we could not provide direct evidence for CYP-induced degradation of the therapeutics. However, we demonstrated a correlation between CYP-activity and response rates across the various cell lines used here. Furthermore, our newly added experiment on the intrinsic G1 phase arrest of the investigated PCa cell lines reveals a correlation between cellular G1 phase arrest and the effectiveness of GE, with the highest G1 arrest rates in VCaP cells, resulting in no therapeutic response to GE.
Moreover, the literature identifies this CYP-variant to have the most significant impact on therapeutic metabolism, further supporting our findings. We have explicitly addressed this limitation (line 271-273) in the discussion section and sincerely appreciate your valuable and insightful feedback on our manuscript.
Comment 4: Insufficient discussion on the limitations of clinical cases: A single case report (PSA level reduction) serves as the starting point of the study, but its small sample size and potential for significant error are not adequately addressed. It is necessary to clarify the preliminary observational nature of this case or supplement the discussion with more clinical research data to support the conclusions.
Response 4: We sincerely appreciate your insightful comment. We aimed to use this case report as a clinical basis for our analysis, although we are fully aware that, due to its unique circumstances (low risk tumor, high fitness, good physical condition and overall health status), this single case should not be used to draw conclusions about PSA reduction solely attributable to GE administration. However, this patient actively reported his experience and draw our attention to potential effects on prostate cancer and potentially relevant drug interactions of GE.
In response to your feedback, we have clearly addressed this important point and explicitly outline the limitations of our case study (line 228-230 and 233-234).
Comment 5: Incomplete statistical methods: The paper does not mention whether multiple comparison corrections (e.g., Tukey HSD) were performed, which may increase the risk of false positives. It is recommended to provide details on the statistical methods, specify the correction methods used, and annotate specific p-values in the figures and tables.
Response 5: We apologize for any confusion regarding our data analysis. Our statistical method throughout the manuscript was one-way analysis of variance (ANOVA). We considered p-values smaller than 0.05 as statistically significant, and we marked intervals with p-values less than 0.05, 0.01, and 0.001 with one, two, and three asterisks, respectively (see section 4.12). No multiple comparison corrections were performed, as our study focused on comparing GE treatment against respective control samples (untreated and ethanol) or pure established medications, we believe that a multiple comparison correction is not strictly necessary for this a priori analysis.
We hope that our methods are now clearer, and we greatly appreciate your helpful feedback.
Comment 6: Optimization needed for figures and terminology: In Figure 5, the scoring criteria for "synergistic/antagonistic effects" (thresholds of 0.8 and 1.6) are not clearly justified; some figures (e.g., immunofluorescence images) have insufficient resolution. It is recommended to provide high-resolution images and ensure clarity in all visual data.
Response 6: In principle, all combination indices CI values greater than 1 are considered antagonistic. However, to enable a more nuanced differentiation among these values, we introduced specific threshold limits. This approach allowed us to further distinguish between the various therapeutic agents and to identify substances, namely, Enzalutamide and Olaparib, that could be further tested in combination and warrant continued observation in preclinical models. We have incorporated this information, along with the corresponding justification, into the Materials and Methods section (line 542-557). We sincerely appreciate your valuable input and hope that this adequately addresses your query on the matter.
We were able to generate different resolutions through the respective experimental setups. In cases where a high resolution was indeed necessary (e.g., Figure 3 analysis of Integrin ß localization), you will see that very high resolution was achieved through specific staining methods, culture surfaces, and microscopy settings. In other instances, where only intensity measurements were required, the necessary resolution parameters were adjusted accordingly.
We have made the original data available, which certainly provide much higher resolution images than those presented in the compiled figures.
We hope that we have now addressed your concerns with these clarifications and sincerely appreciate your helpful comments.
Round 2
Reviewer 2 Report
Comments and Suggestions for Authors
I have made the necessary modifications as per my request